# Impacts of Salt Stress on the Rhizosphere and Endophytic Bacterial Role in Plant Salt Alleviation

Houda Ben Slama [1], Ali Chenari Bouket [2], Faizah N. Alenezi [3], Lenka Luptakova [4], Oleg Baranov [5], Reza Ahadi [6] and Lassaad Belbahri [7,*]

1  NextBiotech, 98 Rue Ali Belhouane, Agareb 3030, Tunisia
2  East Azarbaijan Agricultural and Natural Resources Research and Education Center, Plant Protection Research Department, Agricultural Research, Education and Extension Organization (AREEO), Tabriz 5355179854, Iran
3  Marine Biodiscovery Centre, Department of Chemistry, University of Aberdeen, Aberdeen AB24 3UE, UK
4  Department of Biology and Genetics, University of Veterinary Medicine and Pharmacy in Košice, 04181 Kosice, Slovakia
5  Department of Biological Sciences, National Academy of Sciences of Belarus, 220072 Minsk, Belarus
6  Department of Plant Protection, Faculty of Agriculture, Azarbaijan Shahid Madani University, Tabriz 3751-71379, Iran
7  University Institute of Teacher Education (IUFE), University of Geneva, 24 Rue du Général-Dufour, 1211 Geneva, Switzerland
*  Correspondence: lassaad.belbahri@unige.ch

**Abstract:** Salinity stress is among the key challenges for sustainable food production. It is continuously increasing against the backdrop of constant climate change and anthropogenic practices leading to a huge drop in soil, water, and cultivated crop quality and productivity. Halotolerant plants represent hot spots for endophytic bacteria which may have mechanisms to overcome salt stress. This research initiative aims to highlight the possible exploitation of bacterial endophytes as a microbial biotechnology tool in the productive success of plants exposed to saline stress. We started by solely studying the mechanisms of stress tolerance by plants and halotolerant bacteria. After that, we focused on the beneficial mechanisms of endophytic bacteria in salt stress mitigation. On one side, potent bacterium works by promoting plant performances by facilitating the plant's nutrient uptake (P, K, Zn, N, and Fe) and by promoting the production of growth hormones (IAA and CKs). On the other side, they balance stress phytohormones (ABA, JA, GA, and ACC) produced by plants in case of soil salt augmentation. The selected potent endophytic bacteria could be exploited and applied to ameliorate the production and salt tolerance of food crops. Lastly, we elucidated deeper advanced technologies including (i) genomics unveiling the plant's culture-dependent and culture-independent microbiomes, (ii) metabolomics focusing on genes' metabolic pathways to discover novel secondary metabolites, (iii) transcriptomics studying gene expression, and (iv) proteomics delimiting proteins expressed in stress alleviation. These technologies have been used to understand the plant–bacterial mechanisms of interaction to combat salt stress.

**Keywords:** saline soil; stress mitigation; plant–endophytes interaction; microbial biotechnology; phytohormones; omics





## 1. Introduction

The human population is projected to reach around 10 billion people within the next 30 years [1–3]. This inflation of population threatens global food security and human nutrition, especially since agricultural production is subjected to multiple environmental factors, such as salinity [4], temperature [5], drought [6], presence of toxic metals, and/or organic contaminants among various other stresses [3]. Excess salt concentrations affect 7% of the world's land involving 20% of cultivable fields and approximately 70% of dry land [7,8]. Salt accumulation is increasing constantly owing to anthropogenic practices and

global warming coupled with natural disasters [9–11], leading to a huge drop in soil, water, and cultivated crop quality and productivity [12]. Excess in $Na^+$, $Ca^{2+}$, $Mg^{2+}$, and $SO_4^{2-}$ ions and alkaline pH constitutes a prevalent indicator of a hypersaline environment [13–15]. Soil salinization is often measured by calculating electrical conductivity (EC), and when it exceeds 4 deci-Siemens per meter ($dS \times m^{-1}$) the soil is considered saline [16]. Manishankar et al. [17] cited in their review that, depending on the soil type, the increase in sodium concentrations is responsible for modifying the soil texture by decreasing its dispersion, porosity, and permeability to air, water, and fertilizers [17]. Likewise, depending on the plant species and the growth stage, it may be affected by ionic and osmotic stresses (~200 mM NaCl) sometimes leading to its subsequent mortality [18–21]. The osmotic stress alters the cell's water content; it is triggered by plants immediately after excess NaCl detection. Ionic stress takes place days after NaCl occurrence; it depends on the frequency of $Na^+$ and $Cl^-$ ions accumulated inside the plant cells [22]. Spontaneously growing plants in saline biotopes are known as halophytes and they developed various mechanisms of adaptation to high-salt concentrations (>400 mM) [23,24]. The common processes of soil desalinization by halophytic plants are uptake, accumulation, and/or exclusion of excess salts, maintaining ionic balance using $Na^+/K^+$ transporters, lowering the transpiration rate, hydraulic conductance, and stomatal openings, as well as the expression of genes responsible for salt stress alleviation [25,26]. Thus, to conduct healthy, cost-effective, and biological salt-tolerant agriculture, it is important to use naturally salt-tolerant plants and/or to exploit the plant's associated rhizospheric and endophytic microorganisms in order to enhance endogenous and exogenous plant salt stress tolerance [24,27,28]. These beneficial microbes promote host plant tolerance to soil salinity, increase soil fertility, promote plant growth, and eliminate excess salt in their host plants [29,30]. The mechanisms used by these microorganisms to induce tolerance are osmotic balance, compatible solutes synthesis, exopolysaccharides production, a lipidic layer of Gram-negative bacteria, bacterial consortium interactions, and genetic improvement/modifications of secondary metabolites by regulating the expression of plant stress genes [31–40] as will be further explored throughout this review.

## 2. Effects of Salinity on Plants

Plants subjected to high concentrations of salt show several symptoms at all stages of their growth [41]. The morphological reduction of plant growth rate may be caused by a tremendous decrease in the root and leaf surfaces leading to a deficiency in water and nutrient assimilation and a disturbance in photosynthesis, respectively [42,43]. Halophytic plants have an astonishing ability to cope with extreme hypersaline environments contrary to glycophytes being salt sensitive. Vaishnav et al. [22] stated that most vegetable plants are glycophytes and most cereals and legumes are halophytes (Figure 1).

The plant's response to salt stress starts with discerning stress signals via membrane receptors. These signals act harmonically to alleviate harmful salt concentrations. The study of biochemical responses gives an in-depth knowledge of the plant's immune response to salt stress. For instance, oxidative stress is activated in case of anionic and osmotic damage [44,45]. It induces changes in the plant's physiological response by producing phytohormones, namely ethylene, auxin, cytokinin, gibberellic, and abscisic acids [46,47]. These plant disorders generate high concentrations of ROS (Reactive Oxygen Species) including hydroxyl radicals ($OH^-$), hydrogen peroxide ($H_2O_2$), lipid peroxidation, superoxide ($O^-$), and singlet oxygen ($O_2$) which are responsible for plant cells, proteins, and DNA damages [48–50]. ROS overproduction occurs in mitochondria, chloroplasts, peroxisomes, and apoplast organs [51]. When they exceed the threshold level, ROS become harmful to plant organs, tissues, and principal cell constituents (proteins, lipids, chlorophylls, nucleic acids, etc.) [52,53]. Stressed plants produce antioxidant compounds to lower ROS concentrations by breaking down and eliminating free radicals [54–56]. Enzymatic antioxidants include catalase (CAT), superoxide dismutase (SOD), peroxidases (POX), glutathione reductase (GR), and glutathione transferase (GT) [57–59]. They work in an organized system to

alleviate oxidative damage caused by ROS [60–62]. Other non-enzymatic antioxidants and osmolytes, namely carotenoids, proline, glutathione, ascorbic acid, phenols, flavonoids, and α-tocopherol are involved in ROS scavenging by promoting osmotic balance and preserving the plant's protein structures [63–65]. These resistance mechanisms work together to improve plant adaptation to salt damage [66,67] (Figure 1).

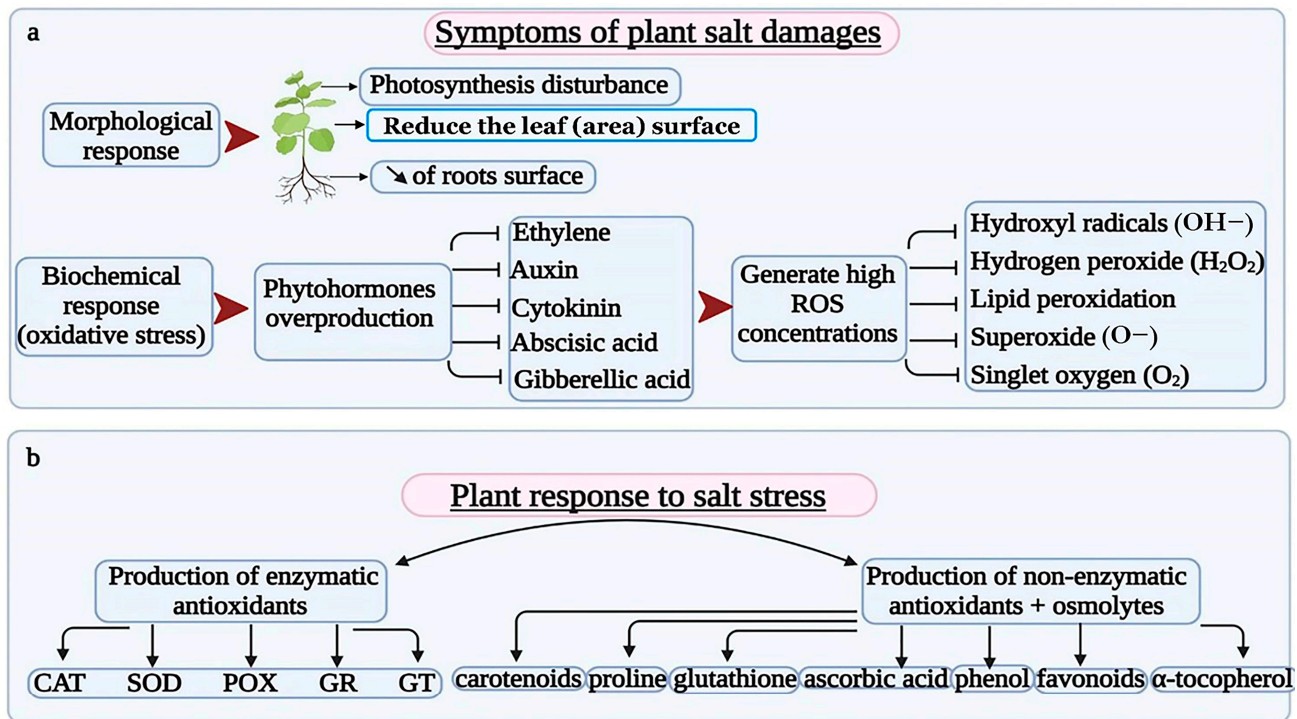

**Figure 1.** Plant salt stress repercussions and mechanisms of salt alleviation. (**a**) symptoms of plant salt damage. (**b**) plant responses to salt stress.

### 3. Effects of Salinity on Plant Microbiomes

Salt stress shapes rhizospheric and resident endophytic microbiota in plants [68]. Therefore, numerous studies have targeted the microbiomes of halophytes and glycophytes revealing a unique microbiome population that is selected by plants grown under salt stress [69,70]. Salinity has also been documented to be among the main factors regulating the bacterial community associated with the roots of halophytes [71]. The microbiomes of plants under salt stress are supposed to provide candidates that can help efforts to counteract salt stress's harmful effects on cultivated crops [70].

### 4. Genes Involved in Plant Protection against Salt

Plant tolerance to salinity has been linked to the presence in their genomes of genes that limit salt uptake by the roots from the soil, limit salt transport throughout the plant, adjust ionic and osmotic balance in plant cells, and regulate the onset of senescence [72]. Therefore, genes that control sodium uptake and transport regulation have been documented in rice [73]. In lentil salt stress, tolerance has been linked to genes that enhance proline accumulation and modulate photosynthetic traits [74]. Numerous genes for salt stress tolerance are now being isolated from halophytes, glycophytes, and plants' wild relatives [70,75,76].

### 5. Bacterial Adaptation to High Salt Levels

Bacteria are endowed with various acclimatization mechanisms to hypersaline conditions [5]. For example, osmotic pressure is a phenomenon occurring between bacterial intracellular and extracellular mediums to maintain an equivalent ion efflux across the

cellular membrane [42,67]. When bacteria absorb extra amounts of salt, $Na^+$ ions remain blocked in the extracellular medium until achieving an osmotic balance [31]. Otherwise, several bacteria could accumulate and/or synthesize compatible solutes including amino acids with their derivatives (proline, betaine, choline, glutamate, etc.), polyols (mannitol, sorbitol, glycerol, etc.), and sugars in huge amounts [32,33]. These compatible solutes do not have a specific charge, and they do not interfere with osmosis. Instead, they act by increasing the volume of cytoplasm and water in bacteria allowing them to grow and tolerate extreme conditions [77–79]. Certain groups of bacteria produce exopolysaccharides (EPS) to form biofilms responsible for maintaining bacterial moisture in case of high NaCl concentrations [36]. Furthermore, the lipid layer existing in the membrane of Gram-negative bacteria helps them improve salinity tolerance [37]. Numerous scientists have also demonstrated that the synergetic interactions in a bacterial consortium are a powerful tool for salt stress mitigation [38]. The genetic improvement/modification of secondary metabolites responsible for salt stress toleration in halotolerant bacteria helps improve their adaptability [2]. Other mechanistic lines of bacterial salt resistance are still not fully understood because endophytic metabolites are constantly changing depending on internal and external conditions [36] (Figure 2).

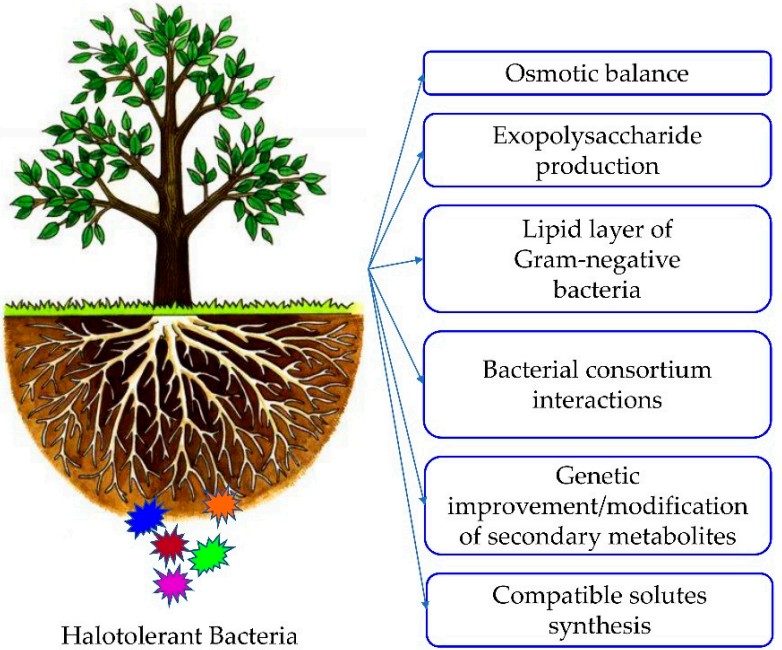

**Figure 2.** Mechanisms of bacterial adaptation to high salt levels.

## 6. Plants–Endophytes–Salt Interactions

Apart from the plant and bacteria's separate mechanisms of salt adaptation, plants could improve their salt stress tolerance through symbiotic associations with rhizospheric and endophytic bacterial communities [23,27,37]. Soil salinity affects rhizosphere communities and creates a natural selection pressure that is positive for halotolerant bacteria [39]. Soil salinity affects rhizospheric communities and creates a positive selection of halotolerant bacteria. This specific group has a significant role in improving both soil and plant health [80].

The rhizosphere is the primary source of endophytes [27], where competitive bacteria are attracted by plant exudates engaging only powerful and beneficial bacteria [39,81]. Bacterial entrance into the host plant occurs either by the secretion of specific enzymes such as cellulase and pectinase which allows them to break down the root cortex [28], or by wounds existing naturally during the propagation of secondary roots and/or caused by phytopathogens [82]. The most common definition of endophytic bacteria includes all those that, during part or all of their life cycle, invade the internal tissues of plants (leaves,

flowers, seeds, stems, and roots) without causing disease and that can confer benefits to their host [83]. For a more detailed definition and history of endophytes, I invite you to read the book chapter in Slama et al. [30]. Shastry et al. [84] reported that the endophytic *Enterobacter cloacae* gene *Ghats I* encodes pectinesterases and cellulases enzymes having a crucial role in bacterial entry inside the host plant's tissues [84].

Rhizospheric and endophytic bacteria positively implicate their associated plants using similar mechanisms [82,85]. However, endophytes have attracted the greatest attention due to their direct interaction with the host plant, less competition with other microbes, and better performance to remove biotic and abiotic stresses [86,87]. In this mutual interaction, plants benefit from bacterial abilities by enhancing nutrient availability and uptake apart from the amelioration of their adaptive and immune systems [48]. Bacteria benefit from available nutrients and protection from external aggressions [29].

Considerable attention has been recently dedicated to applicable projects focusing on the screening of bacterial endophytes as natural biofertilizers and plant protectors [24,28]. In the case of salt stress, it is preferable to use halophytic plants growing in saline environments [88,89]. They constitute the best model for the isolation of bacterial endophytes helping in the elimination of excess salt concentrations [24,90–93]. The isolated halotolerant bacteria could be inoculated to non-halophytic and/or domestic crops to help them cope with excess salt problems [39,94,95]. It is an effective, eco-friendly, economical, and safe approach [96–98]. For instance, the endophytic *Enterobacter* sp. isolated from the halophytic plant *Psoralea corylifolia* L. enhanced the seed vigor index and salt tolerance of the non-host plant *Triticum aestivum* [97]. Sun et al. [37] used the endophytic bacterium *Pantoea alhagi* NX-11 to mitigate salt concentrations of rice seedlings. Similarly, *Pantoea agglomerans* conferred salt stress resistance of durum wheat and improved its growth in such a harsh environment [45] (Figure 3). An *in silico* study conducted by Dąbrowska et al. [98] of the RSH (RelA/SpoT Homologs) Gene family and expression analysis in response to PGPR bacteria and salinity in *Brassica napus* provide a strong basis for future studies targeting plant salt tolerance. The strain *Staphylococcus epidermis* (P-30) proved effective under 20% salt concentration and was endowed with multiple PGP potential [99,100]. In another study, the strain *Brevibacterium linens* RS16 was able to enhance rice salt resistance [101].

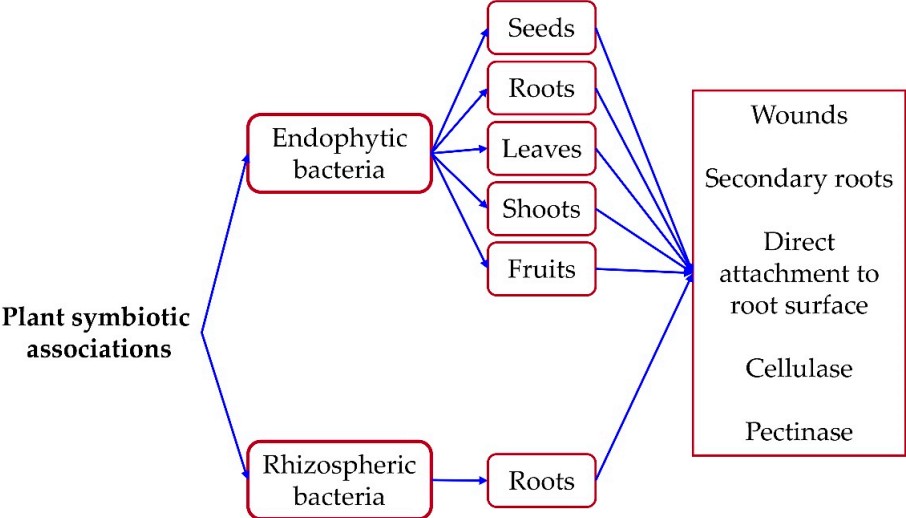

**Figure 3.** Mechanisms of bacterial penetration inside plant tissues.

## 7. Diversification of Endophytic Bacteria Colonizing Halotolerant Plants

### 7.1. Endophytic Lifestyle and Taxonomic Diversification

Endophytic bacterial communities are strongly diversified depending on their lifestyle into obligate and facultative categories [102]. Obligate bacteria are those connected to plants during their entire lifespan. They are related to the host plant in nutrition and they are

transmitted by seeds or vectors only [103]. Facultative bacteria are more autonomous, they could grow inside and outside the host plant without being affected [104]. In most cases, obligate endophytes are culture-independent unlike facultative bacteria being culture-dependent [39].

Saline soil comprises a microbiome belonging to the domains of bacteria, archaea, and eukarya (fungi). Particularly, the domain of bacteria dominates due to its high halotolerant bacterial diversification [105]. The most prevalent bacterial phyla in host plants are Proteobacteria (~50%) comprising the bacterial genera *Pseudomonas* 42P4 and *Enterobacter* 64S1 alleviating tomato saline stress [106], Firmicutes (~10%) such as *Bacillus* sp. isolated from mangrove trees, which improved the percentage of germination and seedling growth when inoculated to rice plants [107] and halotolerant *Bacillus subtilis* Y16 which enhanced *Helianthus annuus* L. to manage salt stress [108], Bacteroidetes (~10%) including the *Flavobacterium crocinum* HYN0056$^T$ species having an upregulation potential of salt and drought stresses of *Arabidopsis* [109], and Actinobacteria (~10%) namely *Streptomyces jiujiangensus* isolated from mangrove trees promote the growth of *Oryza Sativa* seedlings subjected to 200 mM NaCl salinity [110].

### 7.2. Factors Influencing the Bacterial Colonization

The endophytic bacterial diversification is mainly influenced by the host plant's species, physiological status growth stage, and surrounding environment [111]. For instance, Frank et al. [112] illustrated that the endophytic composition varies based on the type of colonized plant tissue, where a remarkable difference could be noticed between species colonizing underground or aerial plant tissues. Plants growing in soil containing high-salt concentrations recruit endophytic bacteria with the capacity to tolerate and alleviate such stress [24]. Borruso et al. [113] conveyed that in harsh environments, the identity of the plant species plays a minor role when compared to the soil contribution in shaping the endophytic communities. However, Szymańska et al. [114] demonstrated that the bacterial composition of a halophytic plant (*Salicornia europaea*) was not affected by high salt levels in the soil.

## 8. Endophytic Bacterial Mechanisms of Salt Mitigation

On the one hand, endophytic bacteria improve crop growth and yield, and, on the other, stimulate plant defense mechanisms [115,116]. The following section will give a better understanding of endophytic bacterial strategies to assist their host plant in enhancing salt mitigation.

### 8.1. Nutrient Uptake Amelioration under Salt Stress

Bacterial mechanisms to ameliorate plant performances at high-salt concentrations involve the contribution of essential nutrient acquisition and the stimulation of plant biomass production. For instance, zinc (Zn), phosphate (P), and potassium (K) exist mostly in insoluble forms in the soil [24,28]. Therefore, bacterial nutrient solubilization is crucial for plant growth [117]. Various bacterial genera ensured P, Zn, and K solubilization. They include *Bacillus*, *Azotobacter*, *Pantoea*, *Pseudomonas*, *Proteus*, *Providencia*, *Serratia*, *Klebsiella*, *Enterobacter*, *Acidothiobacillus*, *Paenibacillus*, and many other genera [15,27,118,119]. Nitrogen (N) is another fundamental element for plants. Biological N-fixation occurs by transforming the atmospheric nitrogen into a plant-assimilable form [7]. *Rhizobium* is the most well-known bacterial genera responsible for N-fixation, and, even in the presence of high NaCl concentrations, it forms nodules in the roots of its host plant where nitrogen is transformed into ammonia using a nitrogenase enzyme [120]. Hanin et al. [121] proved the intervention of halotolerant N-fixing bacteria in preserving membrane ionic balance. Iron predominantly exists in an insoluble form ($Fe^{3+}$) in soil and is implicated in several enzymatic reactions in plants and bacteria [39]. Endophytic bacteria produce siderophores to chelate iron and facilitate its plant uptake [122]. For instance, numerous scientists demonstrated the effectiveness of *Bacillus* genera in iron chelation under a saline environment [123]. Moreover,

Slama et al. [28] conducted a large siderophore screening study on 117 endophytic bacteria isolated from the halophytic plant *Limoniastrum monopetalum*. The results showed that 95% of these bacteria produced siderophores with most of them being of the *Bacillus* sp. species [28] (Figure 4).

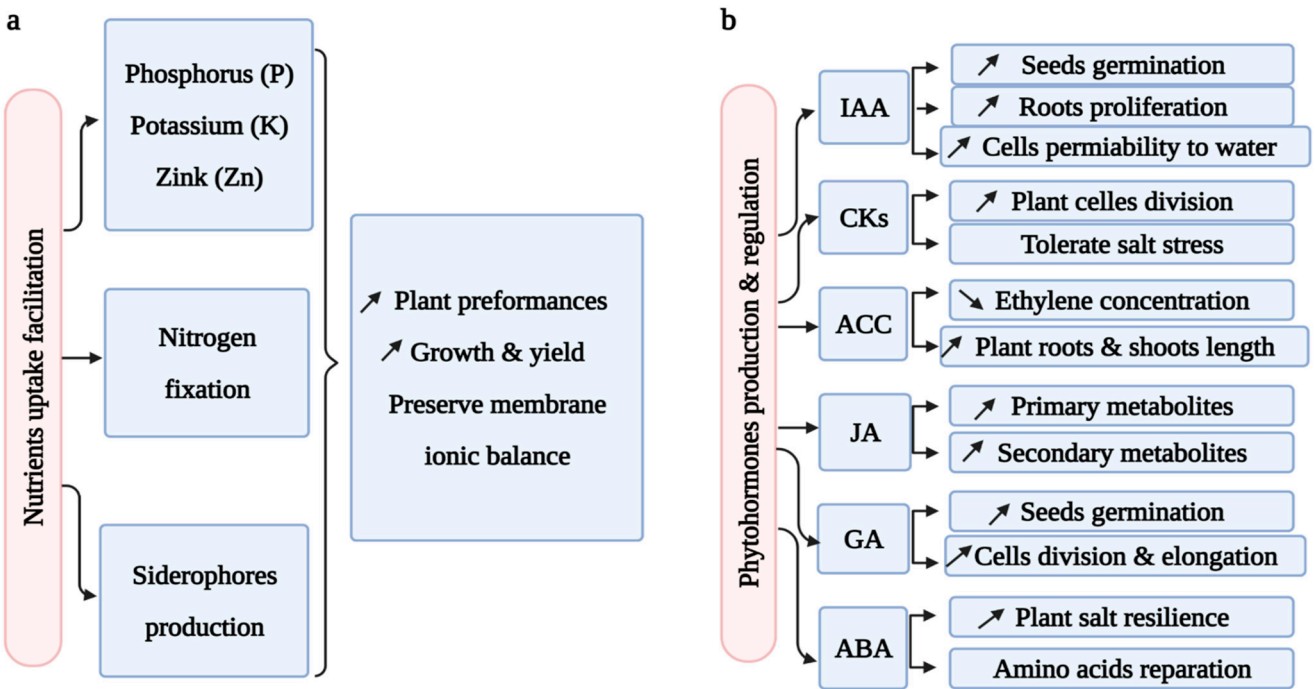

**Figure 4.** Bacterial mechanisms of salt stress tolerance. (**a**) nutrient uptake facilitation. (**b**) phytohormone production and regulation. Upward and downward arrows mean increase and decrease of options.

### 8.2. Phytohormone Production and Regulation under Salt Stress

Phytohormones are organic compounds produced at low concentrations by plants and beneficial microbes to boost plant growth and yield [124]. The modulation of phytohormone levels helps in plant proliferation [125].

Auxin indole-3-acetic acid (IAA) is a significant phytohormone produced by a vast array of endophytic bacterial genera colonizing halophytic plants such as *Marinobacterium*, *Bacillus*, *Sinorhizobium*, *Arthrobacter*, and *Pseudomonas* [126]. In the case of salinity stress, IAA acts by increasing plant seed germination, root proliferation, and cell permeability to water, and by decreasing cell wall pressure [22]. In another study, Soleimani et al. [127] studied the effects of IAA-producing bacteria in improving NaCl toleration. The study revealed that the *Arthrobacter siccitolerans* strain adjusted the IAA expression levels to cope with the salt stress imposed on the host plant.

Cytokinins (CKs) are produced by endophytic bacteria to enhance plant cell divisions and to tolerate environmental stresses including high-salt concentrations [128,129]. Indeed, the plant's inoculation with microbes producing phytohormones including cytokinins helped to inhibit the adverse effects of salt and contributed to the amelioration of all plant growth stages [130].

In the 1-aminocyclopropane-1-carboxylate (ACC) deaminase reaction, S-adenosyl methionine (SAM) is converted by the 1-aminocyclopropane-1-carboxylate synthase (ACS) enzyme to ACC, the immediate precursor of ethylene [131]. Ethylene acts as a plant growth hormone at low concentrations. However, plant stress exposure generates high concentrations of ethylene, transforming it into a plant growth inhibitor which could lead to plant death [132]. Endophytic bacteria could accumulate ACC produced by plants and transform it into ammonia and α-ketobutyrate, leading to a drop in ethylene concentra-

tions, a regulation of plant growth, and protection from saline and any other imposed stresses [133]. Otherwise, ACC is used for nitrogen assimilation in the form of ammonia [39]. A plethora of plant growth-promoting bacteria (PGPB) were able to produce the ACC deaminase enzyme, and they include *Pseudomonas*, *Bacillus*, *Acinetobacter*, *Enterobacter*, *Arthrobacter*, *Serratia*, *Brevibacterium*, *Corynebacterium*, *Planococcus*, *Micrococcus*, *Exiguobacterium*, *Burkholderia*, *Halomonas*, *Zhihengliuella*, *Alcaligenes*, *Ochrobactrum*, *Klebsiella*, and *Oceanimona* genera [134–136]. Singh and Jha [137] demonstrated the great effects of the halotolerant *Serratia* sp. bacterium in lowering the ethylene levels of wheat cultivated in saline soil. This strain enhanced the plant growth rate by facilitating nutrient uptake and increasing wheat roots and shoot length. Yadav et al. [15] reported that *P. simiae* AU5 mutant overproducing ACC deaminase was effective in decreasing salt stress and ethylene concentrations in mung bean plants as compared to the *P. simiae* AU5 wild strain.

Halotolerant bacteria are also able to produce and alter high abscisic acid (ABA) levels in plants in case of abiotic stress occurrence [39,138]. Interestingly, Shahzad et al. [139] stated that the endophytic *Bacillus amyloliquefaciens* strain RWL-1 produced ABA to ameliorate plant salt resilience. It allowed the reparation of essential amino acids and induced salicylic acid production by the host rice plant.

Jasmonic acid (JA) is a signal molecule promoting the production of a plant's primary and secondary metabolites to ameliorate their tolerance to both biotic and abiotic stresses [140]. In previous work, Liu et al. [141] conveyed that *Bacillus amyloliquefaciens* FZB42 conferred salt tolerance to the host *Arabidopsis* plant by upregulating the plant's JA pathways. Moreover, an earlier study found that the bacterial inoculation of plants increases the JA gene expression of host plants [142]. Gibberellic acid (GA) is another plant hormone produced by several endophytes as well. It is responsible for plant cell division and elongation [38], seed germination, and fruit maintenance [143]. PGPB-inoculated plants under salinity stress promote plant growth throughout the production of several phytohormones including GA. For example, the *Pseudomonas putida* H-2-3 strain improved the performances of the host soybean plant cultivated in hypersaline soil [144] (Figure 4).

## 9. Molecular Analysis of Plant–Bacterial Mechanisms of Salt Mitigation

Valuable omics tools (genomics, metagenomics, transcriptomics, and proteomics) have been recently integrated in order to identify and delimit the diversification, roles, and ways of communication within the endophytic bacterial consortium and with their host plants [145]. For instance, genomic identification deeply elucidates the culture-dependent and culture-independent microbiomes in a specific plant, which helps in understanding their mechanisms of action [91,118]. Reportedly, the whole genome of the halotolerant strain *Bacillus fexus* KLBMP 4941 was sequenced to determine the biosynthetic gene clusters (BGCs) involved in salt alleviation [146]. In the same context, metagenomic studies are charged with delimiting the metabolic pathways of genes encoding for known and novel secondary metabolites allowing bacterial adaptation to harsh salinity [147,148]. Similarly, the transcriptomic study acts by revealing genes expressed by bacteria exposed to salt stress [149,150]. This phenomenon was well described by Dong et al. [151] who carried out a transcriptomic analysis on bacteria colonizing *A. thaliana* grown under salt stress. The endophytic bacteria expressed various genes responsible for the salt stress resistance of their host plant. Lastly, the proteomic study gives an insight into the proteins of interest expressed by bacteria for stress mitigation. Genomic sequence and comparative proteomic analysis were realized to determine the response of *Micrococcus luteus* strain SA211 to Lithium (Li) [152–154]. The *M. luteus* strain SA211 was able to adapt by over synthesizing proteins responsible for coping with LiCl stress. Generally, the use of advanced and novel protocols permits the discovery of novel natural products, which could be useful in bacterial adaptation to extreme environments as well as in plant and human health [153]. The above-mentioned high throughput molecular sequencing techniques are extremely valuable to form mutants endowed with high salt resistance capacities and their further biological inoculation in glycophyte plants in order to confer salt tolerance [138] (Figure 5).

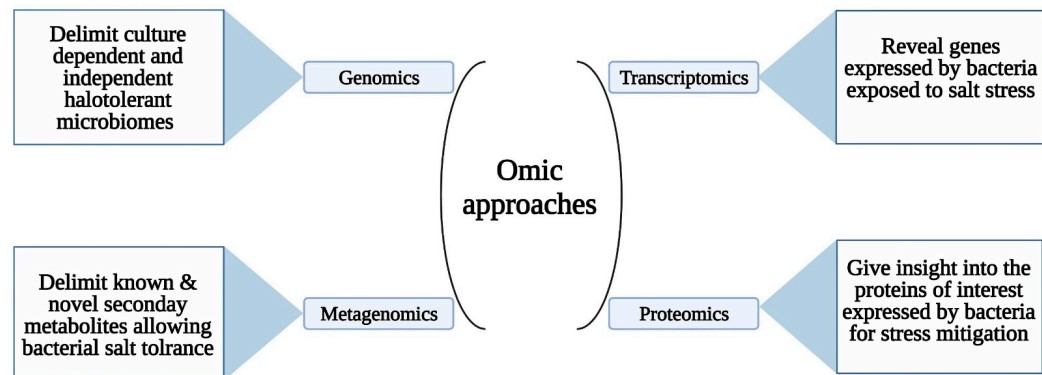

**Figure 5.** Omic approaches enhance knowledge about bacterial mechanisms of salt stress adaptation and mitigation.

## 10. Conclusions

Plant–endophyte interaction is a very complex mechanism controlled by a network of signals, hormones, enzymes, volatile compounds, genes, and metabolites working in tandem to ensure a mutual relationship. Our work described the mechanisms of salt mitigation by plants and bacteria. However, it focused mostly on the advantages provided by endophytic bacteria to improve plant tolerance to saline conditions. We discussed the mechanisms of nutrient uptake amelioration and phytohormone production and the regulation of potent bacteria helping in salt mitigation. Moreover, we highlighted the advanced molecular and omics tools' contribution in revealing the mechanisms of salinity stress. Potent salt-tolerant bacteria isolated from halotolerant plants could be further exploited in agronomy in order to improve agricultural plants' resistance to increasing salt concentrations in soil and water. This biological remediation is easy, cost-effective, and environmentally friendly.

**Author Contributions:** Conceptualization, H.B.S., A.C.B. and L.B.; methodology, L.B.; software, R.A., A.C.B. and H.B.S.; validation, H.B.S., L.L., O.B. and L.B.; formal analysis, O.B., L.L., A.C.B. and L.B.; investigation, H.B.S. and L.B.; resources, F.N.A., O.B. and L.L.; data curation, R.A. and A.C.B.; writing—original draft preparation, H.B.S. and L.B.; writing—review and editing, H.B.S., A.C.B. and L.B.; visualization, F.N.A., H.B.S. and L.L.; supervision, L.B.; project administration, L.B.; funding acquisition, F.N.A., O.B. and L.L. All authors have read and agreed to the published version of the manuscript.

**Funding:** This research received no external funding.

**Institutional Review Board Statement:** Not applicable.

**Informed Consent Statement:** Not applicable.

**Data Availability Statement:** No new data were created in this study.

**Conflicts of Interest:** The authors declare no conflict of interest.

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
