# Peer review of "Impacts of Salt Stress on the Rhizosphere and Endophytic Bacterial Role in Plant Salt Alleviation"

_2037-0164, doi:10.3390/ijpb14020030_

Round 1
Reviewer 1 Report
First I would like to congratulate you for your work. I was very happy to read it, as it is a work that presents a multidisciplinary approach that involves the soil-microorganisms-plant system and the interactions with an abiotic factor that is so worrying for future generations and their food security, due to the increase in arable areas compromised by saline stress.
I made some corrections and suggestions throughout the work, I even recommended some current bibliographies that fit this work, in case you want to add.
Best Regards.

Author Response
Reviewer 1
Open Review
( ) I would not like to sign my review report
(x) I would like to sign my review report
English language and style
( ) English very difficult to understand/incomprehensible
( ) Extensive editing of English language and style required
( ) Moderate English changes required
( ) English language and style are fine/minor spell check required
(x) I don't feel qualified to judge about the English language and style
Is the work a significant contribution to the field? |
|
Is the work well organized and comprehensively described? |
|
Is the work scientifically sound and not misleading? |
|
Are there appropriate and adequate references to related and previous work? |
|
Is the English used correct and readable? |
Comments and Suggestions for Authors
First, I would like to congratulate you for your work. I was very happy to read it, as it is a work that presents a multidisciplinary approach that involves the soil-microorganisms-plant system and the interactions with an abiotic factor that is so worrying for future generations and their food security, due to the increase in arable areas compromised by saline stress.
I made some corrections and suggestions throughout the work, I even recommended some current bibliographies that fit this work, in case you want to add.
Best Regards.
Corresponding author: Thanks for your comment.
Corrections and Suggestions
1 – Line 22 and 23 – In the sentence: “Halotolerant plants represent hot spots for endophytic bacteria with mechanisms allowing salt stress overcoming.” I suggest you change it to a not-so-general phrase, as not all endophytic bacteria have “all” tolerance induction mechanisms, so please change it to: “Halotolerant plants represent hot spots for endophytic bacteria which may have mechanisms to overcome salt stress.”
Corresponding author: It is done.
2 – Line 23 to 25 – I suggest changing the sentence: “This research initiative aims to highlight the possible exploitation of bacterial endophytes to tackle salinity problems on plants.” for: “This research initiative aims to highlight the possible exploitation of bacterial endophytes as a microbial biotechnology tool in the productive success of plants exposed to saline stress.”. In this way, your work will be more relevant for future work and even for their adoption in the inoculation of agricultural plants.
Corresponding author: It is done.
3 – Line 33 – I suggest you change one of the keywords “Nutrients” or “Phytohormones” by the word “Microbial Biotechnology” for your work to stand out in the area of agricultural biotechnology, as your review has a high biotechnological potential.
Corresponding author: It is done.
4 – Line 40 and 41 – The frase: “Excess salt concentrations affect 7% of the world’s land involving 20% of cultivable fields [5,6 ].” It presents very important information. However, I suggest that I read at work - Goswami M, Deka S (2020) Plant growth-promoting rhizobacteria—alleviators of abiotic stresses in soil: A review. Pedosphere, 30(1):40–61. - In it, the authors state that stress influences approximately 70% of dry land in the world.
Corresponding author: It is done.
5 – Lines 63 to 65 – In the sentence: “These beneficial microbes promote their host plant tolerance to soil salinity by enhancing soil fertility, promoting plant growth and eliminating excess salt in their host plants [27,28].” It would be interesting to bring here at least the topics of how bacteria do this, as a preview of what will be presented below, and may even exclude figure 2, which for me is a summary of everything that has been said, although it is not I could understand what the representation of the middle figure is. In this case, I didn't understand if there were different bacteria where there were halotolerant and non-halotolerant ones. If you accept my suggestion, the sentence could stay: “These beneficial microbes promote host plant tolerance to soil salinity, increasing soil fertility, promoting plant growth, and eliminating excess salt in their host plants [27,28]. The mechanisms used by these microorganisms to induce tolerance are: osmotic balance, compatible solutes synthesis, exopolysaccharides production, lipidic layer of gram negative bacteria, bacterial consortium interactions and genetic improvement/ modifications os secondary metabolites [58,59,60,63, 64, 65, 66], as will be further explored throughout this review.”. Probably the numbering of the references will change.
Corresponding author: It is done.
6 – Lines 83 – 85 – As I suggested excluding figure 2, I believe that bringing a figure that shows from the recognition of the signal by the plant to its physiological response right after this explanation would be clearer to the reader.
Corresponding author: It is done.
7 – In Figure 1 – The reduction of the root surface was listed, but not the leaf surface, as described in the text, so it would be interesting to insert an arrow to reduce the leaf area/surface in the aerial part of the plant.
Corresponding author: It is done.
8 – Line 107 – In the sentence “Bacteria are endowed with various acclimatization mechanisms to hypersaline conditions [5].”. It was not clear whether this statement is generalist, that is, for all bacteria, or whether it was referring to rhizospheric and/or endophytic bacteria.
Corresponding author: It is for to rhizospheric and/or endophytic bacteria.
9 – Figure 2 – I suggest excluding it and including it according to observation 5 of my correction.
Corresponding author: It is changed.
10 – Lines 132 and 133 – In the sentence: “Rhizospheric bacteria exist in the rhizospheric soil which is directly attached to the surface of plant roots [68].”. I suggest switching to: “Soil salinity affects rhizosphere communities and creates a natural selection pressure that is positive for halotolerant bacteria [68].”.
Corresponding author: It is done.
11 – Lines 141 to 143 – I believe that at this point in the paper it would be interesting to be more careful and give a more detailed concept, although it is presented later in the work between lines 169 to 174. A good definition of endophytic bacteria are: “Endophytic bacteria are all those that, during part or all of their life cycle, invade the internal tissues of plants (leaves, flowers, seeds, stems and roots) without causing disease and that can confer benefits to their host.”
Corresponding author: It is done.
12 – Lines 182 to 185 – I think it would be interesting to add information from works that evaluated the diversity and richness of endophytic microorganisms in plants, after all, many of them claim that the diversity and richness of endophytic bacteria reduces from the soil to the softest tissues. specialists of the plant in the case of seeds.
I recommend authors read the following book chapter:
Monteiro, P. H. R., da Silva, F. B., de Abreu, C. M., & da Silva, G. J. (2021). Plant growth promoting rhizobacteria in amelioration of abiotic stresses: a functional interplay and prospective. In Plant Growth-Promoting Microbes for Sustainable Biotic and Abiotic Stress Management (pp. 25-49). Springer, Cham.
In this book chapter, the authors address Microbial Biotechnology as a sustainable productive strategy; present examples with several works with several species of rhizospheric and endophytic bacteria that induced tolerance to saline stress in several agricultural cultures; finally used a compilation of information on the proteomics of rhizospheric and endophytic bacteria that induce tolerance to different stressful environments, even saline.
Corresponding author: Thanks to recommendation we amended our paper using this chapter.
Reviewer 2 Report
The article submitted for review concerns an important problem related to the reaction of organisms to salinity. It is a subject widely studied and known, hence the great difficulty in summarizing this information. The review paper entitled "Impacts of salt stress on the ecosystem and endophytic bacterial role in plant salt alleviation" discusses the mechanisms of plant and microorganism reactions to stress caused by salinity.
General remarks:
1. In my opinion, the article does not contain much essential content, and thus does not exhaust the discussed issue.
2. In Chapter 4, more examples of bacteria for which a protective role against salt stress has been demonstrated should be added, e.g. endophytic Serratia liquefaciens and S. plumuthica and their protective role of rapeseed by NaCl. Important genes should be listed in the manuscript, e.g. RSH homologous to the bacterial stringent response, studied in Brassica napus in salinity, as they determine the efficiency of photosynthesis.
3. The article lacks a chapter describing the genes involved in plant protection against salt stress.
4. Since the title of the review paper includes the impact of salinity on the environment, a chapter on changes in the number of microorganisms during plant growth in saline soils should be added.
Detailed notes:
Figure 2 - change the font size in the figure so that all figures are prepared in the same format.
Figure 3 - the figure presents the presented problem in a chaotic way, please correct/change the figure.
Author Response
Reviewer 2
Open Review
( ) I would not like to sign my review report
(x) I would like to sign my review report
English language and style
( ) English very difficult to understand/incomprehensible
( ) Extensive editing of English language and style required
( ) Moderate English changes required
( ) English language and style are fine/minor spell check required
(x) I don't feel qualified to judge about the English language and style
Is the work a significant contribution to the field? |
|
Is the work well organized and comprehensively described? |
|
Is the work scientifically sound and not misleading? |
|
Are there appropriate and adequate references to related and previous work? |
|
Is the English used correct and readable? |
Comments and Suggestions for Authors
The article submitted for review concerns an important problem related to the reaction of organisms to salinity. It is a subject widely studied and known, hence the great difficulty in summarizing this information. The review paper entitled "Impacts of salt stress on the ecosystem and endophytic bacterial role in plant salt alleviation" discusses the mechanisms of plant and microorganism reactions to stress caused by salinity.
General remarks:
- In my opinion, the article does not contain much essential content, and thus does not exhaust the discussed issue.
Corresponding author: After reviewer recommendations and added sections we believe that the manuscript contains now much essential content, and thus exhaust the discussed issue, thanks to reviewer.
- In Chapter 4, more examples of bacteria for which a protective role against salt stress has been demonstrated should be added, e.g. endophytic Serratia liquefaciens and S. plumuthica and their protective role of rapeseed by NaCl. Important genes should be listed in the manuscript, e.g. RSH homologous to the bacterial stringent response, studied in Brassica napus in salinity, as they determine the efficiency of photosynthesis.
Corresponding author: It is done.
- The article lacks a chapter describing the genes involved in plant protection against salt stress.
Corresponding author: The requested section has been added.
- Since the title of the review paper includes the impact of salinity on the environment, a chapter on changes in the number of microorganisms during plant growth in saline soils should be added.
Corresponding author: The requested section has been added.
Detailed notes:
Figure 2 - change the font size in the figure so that all figures are prepared in the same format.
Corresponding author: It is done.
Figure 3 - the figure presents the presented problem in a chaotic way, please correct/change the figure.
Corresponding author: It is done.
Reviewer 3 Report
Overall, this article provides a clear overview of the research initiative and its objectives. The language used is concise and effective in conveying the key information to the reader. The author has provided a good background on the problem of salinity stress and its impact on sustainable food production and highlighted the importance of halotolerant plants and their endophytic bacteria in overcoming this challenge.The methods used in the research initiative are briefly mentioned, including the study of the mechanisms of stress tolerance by plants and halotolerant bacteria, and the focus on the beneficial mechanisms of endophytic bacteria in salt stress mitigation. The results are also briefly summarized, highlighting the various ways in which endophytic bacteria can promote plant performance and balance stress phytohormones.
However, the abstract could benefit from providing more specific details on the advanced technologies used to understand the plant-bacterial mechanisms of interaction to combat salt stress. Additionally, it would be helpful to provide more information on the implications and significance of the research findings for sustainable food production and addressing the challenge of salinity stress. Overall, the abstract provides a good overview of the research initiative, but could benefit from providing more specific details and implications of the research findings.
However, there are a few areas where the article could be improved:
- The article could benefit from more specific references to studies that support the various claims made throughout the text. In some instances, references are provided but not directly linked to the points being made. For example, in the introduction, it is stated that "agricultural production is subjected to multiple environmental factors, such as salinity, drought, temperature, presence of toxic metals and/ or organic contaminants among various other stresses." However, no reference is provided to support this claim. Similarly, in the section on the effects of salinity on plants, it is stated that "plants subjected to high concentrations of salt show several symptoms at all stages of their growth." Again, no specific reference is provided to support this statement.
- The article could benefit from more clarity and specificity in some of the language used. For example, in the introduction, it is stated that "soil salinization is often measured by calculating electrical conductivity, when it exceeds 4 dS×m−1 the soil is considered saline." This sentence could be clarified by specifying what "dS×m−1" means and providing a reference to support this threshold value. Similarly, in the section on the mechanisms of salt tolerance in halophytes and associated microorganisms, it is stated that these organisms use "genetic improvement/modifications of secondary metabolites" to induce tolerance. This phrase could be clarified by specifying what "genetic improvement/modifications" refers to and providing a reference to support this claim.
- The article could benefit from more discussion of the practical applications of the research described. While the article does touch on the potential benefits of using salt-tolerant plants and associated microorganisms in agriculture, there is little discussion of how these practices could be implemented in practice. Providing more specific examples of how these practices are currently being used or could be used in the future would help to make the article more informative and practical for readers.
Overall, this is a well-written and informative article that could be improved with more specific references, more clarity in language, and more discussion of practical applications.
Author Response
Overall, this article provides a clear overview of the research initiative and its objectives. The language used is concise and effective in conveying the key information to the reader. The author has provided a good background on the problem of salinity stress and its impact on sustainable food production and highlighted the importance of halotolerant plants and their endophytic bacteria in overcoming this challenge. The methods used in the research initiative are briefly mentioned, including the study of the mechanisms of stress tolerance by plants and halotolerant bacteria, and the focus on the beneficial mechanisms of endophytic bacteria in salt stress mitigation. The results are also briefly summarized, highlighting the various ways in which endophytic bacteria can promote plant performance and balance stress phytohormones.
However, the abstract could benefit from providing more specific details on the advanced technologies used to understand the plant-bacterial mechanisms of interaction to combat salt stress. Additionally, it would be helpful to provide more information on the implications and significance of the research findings for sustainable food production and addressing the challenge of salinity stress. Overall, the abstract provides a good overview of the research initiative, but could benefit from providing more specific details and implications of the research findings.
Corresponding Author: Thank you for your valuable comments. Required information were added to the “abstract”.
However, there are a few areas where the article could be improved:
- The article could benefit from more specific references to studies that support the various claims made throughout the text. In some instances, references are provided but not directly linked to the points being made. For example, in the introduction, it is stated that "agricultural production is subjected to multiple environmental factors, such as salinity, drought, temperature, presence of toxic metals and/ or organic contaminants among various other stresses." However, no reference is provided to support this claim. Similarly, in the section on the effects of salinity on plants, it is stated that "plants subjected to high concentrations of salt show several symptoms at all stages of their growth." Again, no specific reference is provided to support this statement.
Corresponding Author: Thank you for your valuable comments. Specific references were added in both the “introduction” and in the “effects of salinity on plants section”.
- The article could benefit from more clarity and specificity in some of the language used. For example, in the introduction, it is stated that "soil salinization is often measured by calculating electrical conductivity, when it exceeds 4 dS×m−1 the soil is considered saline." This sentence could be clarified by specifying what "dS×m−1" means and providing a reference to support this threshold value. Similarly, in the section on the mechanisms of salt tolerance in halophytes and associated microorganisms, it is stated that these organisms use "genetic improvement/modifications of secondary metabolites" to induce tolerance. This phrase could be clarified by specifying what "genetic improvement/modifications" refers to and providing a reference to support this claim.
Corresponding Author: Thank you for your valuable comments. Necessary explanations were executed.
- The article could benefit from more discussion of the practical applications of the research described. While the article does touch on the potential benefits of using salt-tolerant plants and associated microorganisms in agriculture, there is little discussion of how these practices could be implemented in practice. Providing more specific examples of how these practices are currently being used or could be used in the future would help to make the article more informative and practical for readers.
Corresponding Author: Thank you for your valuable comments. I have explained in the manuscript that salt-tolerant plants are basically used for the isolation of halophytic bacteria. These associated microorganisms could be inoculated to non_halophytic and/ or domestic crops to help them cope with excess salt problems and promote their growth as explained in “plant-endophytes- salt interactions” section. I added some examples of bacterial microorganisms contribution in plant growth promotion and salt mitigation as recommended.
Overall, this is a well-written and informative article that could be improved with more specific references, more clarity in language, and more discussion of practical applications.
Reviewer 4 Report
Review of the article: “Impacts of salt stress on the ecosystem and endophytic bacterial role in plant salt alleviation”
This article shows an analysis of the endophytic bacteria in plants, it’s a basic review since each subtitle was analyzed superficially. As an example, Omics has not described the results of the genes involved in the ability of the bacteria to live as endophytic or from the plant as well.
Title: It is better to change the word ecosystem for rhizosphere
Line 85; Please describe what it means ROS
Line 125; Which kind of halotolerant bacteria is in the communities of the rhizosphere?
Lines 159 & 160; You must say something about the history and definition of the endophytes since this article talks about it
Line 182; Erase the word Elegant, since molecular analysis has been used since a long time ago
Line 199; It will be nice to say which species of each phylum are in the microbiome of saline soil
Line 294; Change the title for Molecular analysis of ……
Line 311; Start the sentence as Genomic sequence ……
Line 313; I don’t understand the sentence SA211 was able…..
Line 315; change the word gateways and write again the whole sentence; per example; Generally the use of new protocols….
329-330; Write again the sentence; Advanced…..

Author Response
Review of the article: “Impacts of salt stress on the ecosystem and endophytic bacterial role in plant salt alleviation”
This article shows an analysis of the endophytic bacteria in plants, it’s a basic review since each subtitle was analyzed superficially. As an example, Omics has not described the results of the genes involved in the ability of the bacteria to live as endophytic or from the plant as well.
Corresponding Author: Thank you for your valuable comment. We focused on omics in salt stress mitigation and I have added an explanation of genes involved in the ability of bacteria to live as endophyte inside the host plant in the “plant-endophytes-salt interaction” section.
Title: It is better to change the word ecosystem for rhizosphere
Corresponding Author: Thank you for your valuable comment. It is done.
Line 85; Please describe what it means ROS
Corresponding Author: Thank you for your valuable comment. ROS was explained as Reactive Oxygen Species.
Line 125; Which kind of halotolerant bacteria is in the communities of the rhizosphere?
Corresponding Author: Thank you for your valuable comment. I have mentioned the most existing bacterial phyla with examples of species in the “endophytic lifestyle and taxonomic diversification section”.
Lines 159 & 160; You must say something about the history and definition of the endophytes since this article talks about it
Corresponding Author: Thank you for your valuable comment. Necessary explanations were added.
Line 182; Erase the word Elegant, since molecular analysis has been used since a long time ago
Corresponding Author: Thank you for your valuable comment. It is done.
Line 199; It will be nice to say which species of each phylum are in the microbiome of saline soil
Corresponding Author: Thank you for your valuable comment. Examples of species of each phylum were added in the section “Endophytic lifestyle and taxonomic diversification”.
Line 294; Change the title for Molecular analysis of ……
Corresponding Author: Thank you for your valuable comment. It is done in to the text as “Molecular analysis of plant-bacterial mechanisms of salt mitigation”
Line 311; Start the sentence as Genomic sequence ……
Corresponding Author: Thank you for your valuable comment. It is done.
Line 313; I don’t understand the sentence SA211 was able…..
Corresponding Author: Thank you for your valuable comment. It is done in to the text as “M. luteus strain SA211”
Line 315; change the word gateways and write again the whole sentence; per example; Generally the use of new protocols….
Corresponding Author: Thank you for your valuable comment. The sentence was corrected.
329-330; Write again the sentence; Advanced…..
Corresponding Author: Thank you for your valuable comment. The sentence was corrected.
Round 2
Reviewer 1 Report
Dear authors,
I reviewed the corrections changes and suggestions and I liked it very much. I believe the new approach has made the work easier to understand.
I believe that this work will greatly collaborate with research that permeates microbial biotechnology and will instigate new research involving bacteria that have the ability to induce tolerance to saline environments.
Yours sincerely.
Author Response
Dear authors,
I reviewed the corrections changes and suggestions and I liked it very much. I believe the new approach has made the work easier to understand.
I believe that this work will greatly collaborate with research that permeates microbial biotechnology and will instigate new research involving bacteria that have the ability to induce tolerance to saline environments.
Yours sincerely.
Corresponding Author: We appreciate you for your efforts and help to improve the paper.